# Unleashing the Biological Potential of *Fomes fomentarius* via Dry and Wet Milling

**DOI:** 10.3390/antiox10020303

**Published:** 2021-02-16

**Authors:** Abdul Karim Darkal, Mhd Mouayad Zuraik, Yannick Ney, Muhmmad Jawad Nasim, Claus Jacob

**Affiliations:** Division of Bioorganic Chemistry, School of Pharmacy, Saarland University, 66123 Saarbruecken, Germany; abda00001@stud.uni-saarland.de (A.K.D.); s8mhzura@stud.uni-saarland.de (M.M.Z.); yannick.ney@uni-saarland.de (Y.N.); jawad.nasim@uni-saarland.de (M.J.N.)

**Keywords:** antioxidant, ball milling, electrochemistry, *Fomes fomentarius*, heat sterilization, phytochemical screening, redox sponge

## Abstract

*Fomes fomentarius*, usually referred to as tinder conk, is a common wood-based fungus rich in many interesting phytochemicals and with an unique porous structure. Dry or wet ball milling of this sponge on a planetary mill results in small particles with sizes in the range of 10 µm or below. Suspended in water and without preservatives or other stabilizers, the resulting micro-suspensions are sterile for around six days, probably due to the increased temperatures of around 80 °C especially during the wet milling process. The suspensions also exhibit excellent antioxidant activities as determined in the DPPH, ferric reducing antioxidant potential (FRAP) and 2,2′-azino-*bis*(3-ethylbenzothiazoline-6-sulfonic acid (ABTS) assays. In the DPPH assay, IC_50_ values of 0.02–0.04% *w*/*v* and 0.04% *w*/*v* were observed for dry and wet milled samples, respectively. In the FRAP assay, IC_50_ values of <0.02% *w*/*v* and 0.04% *w*/*v* were observed for dry and wet milled samples, respectively. In contrast, the ABTS assay provided IC_50_ values of 0.04% *w*/*v* and 0.005% *w*/*v*, respectively. Notably, this activity is mostly—albeit not exclusively—associated with the highly porous particles and their large surfaces, although some active ingredients also diffuse into the surrounding aqueous medium. Such suspensions of natural particles carrying otherwise insoluble antioxidants on their surfaces provide an interesting avenue to unleash the antioxidant potential of materials such as sponges and barks. As dry milling also enables longer storage and transport, applications in the fields of medicine, nutrition, agriculture, materials and cosmetics are feasible.

## 1. Introduction

Antioxidants represent a class of interesting and often quite reactive and reductive substances which have attracted considerable interest in medicine, nutrition, agriculture, materials, food processing and cosmetics. Natural antioxidants, such as secondary metabolites found in many plants, fungi and (micro-)organisms are especially attractive, yet also often insoluble in water and difficult to extract, store, purify and to deliver [1,2]. Indeed, many natural sources rich in such natural antioxidants, such as barks, roots, seeds and leaves, are difficult and cumbersome to process, with issues ranging from costs and the need for solvents to limited bioavailability and, quite trivially, the risk of fouling and rotting of such materials [2]. *Fomes fomentarius*, the tinder conk, is a sponge found worldwide as a parasite on various trees and is rich in various interesting substances [3]. In theory, the fruiting bodies of the conk can be harvested easily and with good yield, as depicted in Figure 1. The fruiting body is quite easy to handle, store, transport and process and as it is also rich in many interesting redox active and antioxidant substances, *Fomes fomentarius* should therefore provide a fine source of powerful antioxidants, such as the *ortho*-quinone 3,4-dihydrobenzaldehyde (3,4-DHBA) and the gallic acid derivative purpurogallin [4,5,6,7]. Nonetheless, extraction of such (redox-) active ingredients requires (a) solvents such as ethanol and (b) elevated temperatures and the process itself (c) is tedious and (d) is not possible in some instances such as for polymeric glucans and chitin. Moreover, the resulting extracts, such as the ethanolic extract, are not always stable and also not readily useable in practice.

As the tinder sponge in its dried form is very hard and brittle, milling the entire material to micro- and nanoparticles with a planetary ball mill and subsequently suspending these particles may therefore provide an attractive alternative to solvent extraction. This approach towards micro- and nanosizing entire plant samples has been explored by us recently with some success [8,9,10]. Here we have applied ball milling to the tinder sponge as this fungus is amenable to milling rather than extraction, as a sponge provides a very large surface area already, is rich in many interesting yet insoluble ingredients and thanks to its abundance could be used and applied widely in practice [11,12,13].

## 2. Materials and Methods

### 2.1. Reagents and Culture Media

#### 2.1.1. Molisch’s Reagent

The reagent was prepared by dissolving 15 g of *α*-naphthol in 100 mL of ethanol.

#### 2.1.2. Fehling’s Solutions

Fehling’s solution A was prepared by dissolving 0.73 g of copper sulfate pentahydrate (CuSO_4_ 5H_2_O) in 100 mL distilled water and Fehling’s solution B was prepared by mixing 34.6 g of potassium sodium tartrate tetrahydrate (C_4_H_4_KNaO_6_ 4H_2_O) and 10 g of NaOH in 100 mL of distilled water.

#### 2.1.3. Benedict’s Reagent

The reagent was prepared by dissolving 17.3 g of sodium citrate (C_6_H_5_Na_3_O_7_) and 10 g of sodium carbonate (Na_2_CO_3_), in 85 mL of distilled water to form a carbonate citrate buffer followed by the addition of a solution of CuSO_4_ 5H_2_O (1.73 g in 10 mL of distilled water) and the final volume was subsequently adjusted to 100 mL with distilled water.

#### 2.1.4. Wagner’s Reagent

The reagent was prepared by dissolving 2 g of iodine (I_2_) and 6 g of potassium iodide (KI) in 100 mL of distilled water.

#### 2.1.5. Dragendorff’s Reagent

The reagent was prepared by dissolving 0.85 g of bismuth oxynitrate (Bi_5_H_9_N_4_O_22_) in 10 mL of glacial acetic acid and 40 mL of distilled water, followed by the addition of 50 mL of 50% aqueous KI solution.

#### 2.1.6. Mayer’s Reagent

The reagent was prepared by dissolving 1.36 g of mercuric chloride (HgCl_2_) in 60 mL distilled water followed by the addition of a solution of KI (5 g in 10 mL distilled water) and the final volume was adjusted to 100 mL with distilled water.

#### 2.1.7. Phosphate Buffer Saline (PBS)

PBS solution was prepared by mixing 4 g of sodium chloride (NaCl), 0.2 g of potassium chloride (KCl), 0.72 g of disodium hydrogen phosphate (Na_2_HPO_4_) and 0.12 g of potassium dihydrogen phosphate (KH_2_PO_4_) into 500 mL of distilled water. The pH of the buffer was adjusted to 7.4 with HCl (0.1 M) and NaOH (5 M) followed by sterilization via autoclave. The buffer was stored at 4 °C.

#### 2.1.8. Yeast Peptone Dextrose (YPD) Medium

YPD medium was prepared by dissolving 10 g of Bacto peptone, 5 g of yeast extract and 10 g of D-glucose monophosphate in 500 mL of distilled water. The pH of the medium was adjusted to pH 7.2 with HCl (0.1 M) and NaOH (5 M) followed by sterilization via autoclave. The medium was stored at 4 °C.

#### 2.1.9. Lysogeny Broth (LB) Preparation

LB medium was prepared by dissolving 5 g of Bacto peptone, 2.5 g of yeast extract and 5 g of NaCl in 500 mL of distilled water. The pH of the medium was adjusted to pH 7.5 with HCl (0.1 M) and NaOH (5 M) followed by sterilization via autoclave. The medium was stored at 4 °C.

### 2.2. Collection of Fomes fomentarius and Identification

Samples of fruiting bodies of the tinder sponge were collected in the woodlands near the town of Hassel/Saar in Saarland, Germany at Latitude 49.273856°, Longitude 7.173341°, as illustrated in Figure 1. Specimens were removed intact and with a chisel from lime and oak trees and sent to the RWTH Aachen, Aachen, Germany where the specimens were identified as *Fomes fomentarius* by Dr Martin C. H. Gruhlke from the Department of Plant Physiology. In parallel, freshly harvested samples were dried at 40 °C until a constant weight was achieved, which usually took around ten days, and subsequently stored in a dry place until further use. Please note that no trees were harmed during the harvest of *Fomes fomentarius*.

### 2.3. Crushing and Milling

The dried samples were divided into smaller pieces with a household hammer and subsequently crushed employing a household flour mill “Komo Fidibus Medium” (KoMo GmbH & Co. KG, Hopfgarten, Tyrol, Austria). The coarsely powdered fine brown flour obtained is shown in Figure 2 and was subsequently milled employing a planetary mill “Pulversisette 7 premium line” (Fritsch GmbH, Idar-Oberstein, Rhineland Palatinate, Germany).

The dry-milled samples were prepared by introducing 500 mg of the ground powder and 60 g of zirconium oxide (ZrO_2_) milling balls (1 mm in diameter) inside the operating beaker of the planetary mill and operating it continuously at 950 rpm for 5 min. The process resulted in a temperature of 41.6 °C inside the operating beaker. After milling the sample was filtered through a 40 μm sieve.

The wet-milled sample was prepared by adding 500 mg of the coarse milled sample with 60 g of zirconium oxide (ZrO_2_) milling balls (1 mm of diameter) and 40 mL of sterile distilled water. The planetary mill was operated continuously at 1050 rpm for 40 min, resulting in a temperature of 79.1 °C inside the operating beaker. After completion, ZrO_2_ balls were separated and the suspension containing 1.25% *w*/*v* of the wet-milled sample was decanted into sterile Falcon tubes until further use.

### 2.4. Characterization of Samples

Samples were analyzed for size distribution employing Laser Diffraction (LD, also referred to as Static Light Scattering) with the aid of a Mastersizer Malvern 2000 (Malvern Instruments, Worcestershire, UK) attached to a HydroMV dispersion unit at room temperature. Mie theory was applied by utilizing the optical parameter (real/imaginary refractive index) 1.570/0.010.

In order to assess possible microbial contaminations, a few mg of dried milled sample were suspended in 1 mL of sterilized distilled water to prepare a suspension. An amount of 10 μL from both dry and wet milled suspensions, respectively, were each added into 20 mL of YPD and LB medium. Medium was employed as negative control whilst medium containing *Saccharomyces cerevisiae* in YPD medium and *Escherichia coli* in LB medium served as positive controls. The optical densities of the samples were recorded at a wavelength of 600 nm on a Cary50 Bio UV/VIS spectrophotometer (Varian Australia Pty Ltd., Mulgrave, Australia) at a specific time every day for seven days. The experiments were carried out in triplicate.

### 2.5. Ethanolic Extract

An amount of 50 g of the coarse milled sample was suspended in 1.5 L of ethanol (98% *v*/*v*) and stirred for seven days at room temperature followed by filtration. The filtrate was subsequently evaporated employing a rotatory evaporator (BUCHI Labortechnik GmbH, Essen, Germany) to yield 2.07 g of the extract.

### 2.6. Preparation of Suspensions, Supernatants and Pellets

The suspension of dry milled sample was prepared by suspending 15 mg of the milled sponge in 15 mL of sterile distilled water. The concentration was subsequently adjusted according to the assay. For each of the assays, the suspension was sonicated just before use. In order to distinguish between activities associated with the material and activities associated with substances liberated into the aqueous medium, suspensions were centrifuged at 19,000× *g* for 10 min and the supernatant, usually around 4 mL, was collected in a separate tube, whilst the pellet was re-suspended in 4 mL of sterile distilled water for further investigation.

### 2.7. Antioxidant Assays

#### 2.7.1. 2,2-Diphenyl-1-Picryl-Hydrazyl-Hydrate (DPPH) Capacity

The DPPH assay was carried out according to the protocol described in the literature [8,15]. A 0.2 mM solution of DPPH was prepared by dissolving 7.89 mg of DPPH in 100 mL of methanol (98% *v*/*v*) and the resulting solution was kept for 1 h at room temperature. An amount of 200 µL of different concentrations of samples, extracts and references were added to 1 mL of DPPH solution and 800 µL of Tris buffer (0.77 M, pH 7.4) in order to initiate the reaction and the mixture was then incubated at 37 °C for 30 min. Solutions comprised of 800 μL of Tris buffer, 1 mL of sterile distilled water and 200 μL of suspensions and extracts in the absence of DPPH were employed as blanks for the respective samples and a solution comprising of 800 μL of Tris buffer in 1200 μL of sterile distilled water was used as a blank for ascorbic acid. DPPH solution without and with ascorbic acid were employed as negative and positive controls, respectively. The assay was performed in triplicate (*n* = 3) and the radical scavenging activities were observed spectrophotometrically at the wavelength of 517 nm.

#### 2.7.2. Ferric Reducing Antioxidant Potential (FRAP) Capacity

The FRAP assay was carried out according to the protocol described in the literature [8,16]. A stock solution of FRAP (50 μg·mL^−1^) was prepared by mixing 1 mL of ferric chloride (FeCl_3_, 20 mM) aqueous solution, 1 mL of tripyridyltriazine (10 mM) solution prepared in hydrochloric acid (40 mM) and 10 mL of acetate buffer (300 mM, pH 3.6). Different concentrations of samples, extracts, references and standards were added to the solution in order to start the reaction. Samples of suspensions and extracts excluding FRAP were used as blank for the respective samples whilst acetate buffer (300 mM, pH 3.6) was employed as a blank for ascorbic acid. FRAP solution without and with ascorbic acid were employed as negative and positive controls, respectively and FeSO_4_.solution (1 mM) was utilized as the standard. The assay was performed in triplicate (*n* = 3) and the antioxidant activities were observed spectrophotometrically at the wavelength of 593 nm.

#### 2.7.3. 2,2′-Azino-*bis*(3-Ethylbenzothiazoline-6-Sulfonic Acid (ABTS) Capacity

The ABTS assay was carried out according to the protocol described in the literature [8]. A 7 mM aqueous solution of ABTS was reacted with a 2.4 mM aqueous solution of potassium persulfate (K_2_S_2_O_8_) and the resulting mixture was kept in the dark for 16 h in order to generate ABTS^●+^ radicals. The assay solution was subsequently diluted with ethanol in order to achieve an absorbance value of 0.70 ± 0.02 at the wavelength of 734 nm. Different concentrations of samples, extract and references were added to the solution of ABTS in order to start the assay reaction. Samples of suspensions and extracts excluding ABTS solution were employed as blank for the respective samples and sterile distilled water was used as a blank for ascorbic acid. ABTS solution without and with ascorbic acid were utilized as negative and positive controls, respectively. The assay was performed in triplicate (*n* = 3) and the free radical scavenging activities were determined spectrophotometrically at the wavelength of 734 nm.

### 2.8. Electrochemical Analysis

In order to obtain further information about the redox properties of wet milled suspension and ethanolic extract, the electrochemical method of Cyclic Voltammetry (CV) was employed. The suspensions at concentrations of 0.08% *w*/*v* and benchmark compounds relevant to this study, namely 3,4-DHBA and purpurogallin, at a concentration of 0.08% *w*/*v*, were investigated in magnetically stirred 0.1 M sodium phosphate buffer (pH = 7.4), which also acts as supporting electrolyte, with a BAS 100 B potentiostat (BASi West Lafayette, IN, USA) and a three-electrode system involving a glassy carbon working electrode (1.6 mm in diameter), a platinum wire counter electrode and a 3 M KCl Ag|AgCl reference electrode stored in 3 M KCl prior to use. Cyclic Voltammogramms (CVs) were recorded at a scan rate of 0.1 V s^−1^ at 25 °C in three different independent repetitions (*n* = 3), with extensive polishing of the working electrode on cloth pads (MF-1040, BASi, West Lafayette, IN, USA) soaked with an Al_2_O_3_ suspension (MicroPolish Alumina, 0.05 μm particles, Buehler, Lake Bluff, IL, USA) followed by rinsing with distilled water.

### 2.9. Qualitative Screening for Reactive Substances

In order to detect reactive and especially reductive substances in the suspensions and extracts, different qualitative tests were applied, namely the Dragendorff’s, Mayer’s and Wagner’s tests for the presence of alkaloids, Benedict’s, Fehling’s and Molisch’s tests for carbohydrates, the foam test for saponin glycosides, Keller-Killiani and Legal’s tests for cardiac glycosides, and the Salkowski’s test for steroids. The content of phenolic acids was quantified employing the protocol described in the literature [17,18,19,20].

### 2.10. Nematode Assay

The nematode assay based on the agricultural nematode *Steinernema feltiae* was performed as described in the literature [8,10]. Nematodes were purchased from Sautter und Stepper GmbH (Ammerbuch, Baden-Wuerttemberg, Germany) and stored at 4 °C in the dark. 200 mg of *Steinernema feltiae* were suspended in 50 mL of phosphate buffered saline solution (0.1 M, pH = 7.4 ± 0.1) and stirred for 30 min. In order to analyze the quality of nematodes, an initial viability assay was performed before each assay and nematode samples with a viability of over 90% were employed for further experiments. Samples were added to the suspension of nematodes and the initial viability was determined, with a second recording after 24 h and viability was expressed as a percentage value. The buffer and 70% ethanol were employed as negative and positive controls, respectively. The assay was performed in triplicate on three different occasions (*n* = 9).

### 2.11. Statistical Analysis

Data such as antioxidant capacity and nematicidal activities were expressed as means ± SD. For nematicidal activity, data comparisons were performed using One-way analysis of variance (ANOVA), and post hoc analysis was carried out by the Student Newman-Keuls (SNK) test. GraphPad Prism (Version 5.03, GraphPad Software, San Diego, CA, USA) was used for data analysis and to produce charts. A value of *p* < 0.05 is considered statistically significant.

## 3. Results

Several samples of *Fomes fomentarius* were subjected to intense dry and wet milling with a planetary ball mill, resulting in fine powders and suspensions. An analysis of the resulting suspensions shows that the particles have diameters of around 10 µm, and that these particles exhibit considerable antioxidant activity, most likely due to redox events at the vast surface areas of these extraordinarily porous materials. These findings will be presented further and discussed in more detail.

### 3.1. Dry and Wet Milling

In contrast to other conventional methods of nanosizing, such as high pressure homogenization techniques, milling in planetary ball mills provides several advantages, as size reduction may be performed in one step, at elevated temperatures and, especially, under wet and also under dry conditions. Indeed, after drying the sponge in an oven at 40 °C for ten days, manually crushing samples with a household hammer and powderizing them to millimeter size in a household flour mill, dry milling was the preferred method of choice, resulting in a very fine powderous material of brown color, as shown in Figure 2. Like wet milling in sterile distilled water, dry milling with zirconium oxide 1 mm milling balls at a speed of 950 rpm for 5 min resulted in fairly uniform particles with diameters in the range of 1 to 10 µm, as indicated in Figure 3. Notably, this size reduction in the planetary ball mill to the micrometer size was very reproducible and yielded similar particles under dry and wet milling conditions, with the notable difference that the dry milling produced a fine powder which could be stored and suspended in water on demand, whereas the wet milled sample was suspended already and its handling and storage are therefore more immediate and difficult.

In any case, the resulting particle suspensions, in the absence of any stabilizers or preservatives, tend to sediment within 5 min, and therefore require agitation before application. Whereas this may be considered as a possible shortcoming of ball milling when compared to other methods of nanosizing, there are also some advantages. Ball milling is a highly exothermic process and temperatures in the chamber can reach easily 80 °C. The resulting powders and suspensions, respectively, thus often were sterilized automatically by the milling process and turned out to be rather resistant to microbial contamination, with no significant fouling for up to six days, as illustrated in Figure 3.

### 3.2. Antioxidant Activity

Based on the composition of such conks reported in the literature, which includes a number of redox active and reducing substances such as glucans, chitin, melatonin, hemicellulose and glucuronic acid, a considerable antioxidant activity associated with this natural material may be expected [21,22,23,24]. Antioxidant activity therefore was determined by three independent common antioxidant assays, namely the DPPH, FRAP and ABTS assays, which are indicative of different aspects and reactivities of antioxidants and antioxidant and radical scavenging activities.

#### 3.2.1. DPPH Capacity

Among the various antioxidant assays, the DPPH assay is employed frequently to determine radical scavenging activity. As expected, significant concentration dependent activities of the dry milled and wet milled samples were observed in this assay, as shown in Figure 4. In the case of dry milled *Fomes fomentarius* suspended in sterile distilled water, the resulting suspension at a concentration of just 0.16% *w*/*v* sequestered 86% of the DPPH radicals, with an estimated IC_50_ value in the range of 0.02 to 0.04% *w*/*v* as shown in Figure 4a. Interestingly, this pronounced radical scavenging activity of the suspension in freshly prepared samples at first can be assigned to the microparticles themselves, rather than to the surrounding solution, as separation of the particles from the supernatant and subsequent resuspension of the particles demonstrated that the supernatant is considerably less active than the particles, with around 48% radical sequestration by the supernatant and over 80% by the resuspended particles, as highlighted in Figure 4b and c, respectively. After stirring the suspension (0.08% *w*/*v*) at room temperature for 24 h, this radical sequestering activity increases in the supernatant from 26% to around 74% and decreases in the particle fraction from 73% to about 43%, indicating a slow liberation of soluble antioxidants from the particles into the supernatant, as shown in Appendix A.

In the case of the wet milled sample, a similar activity was observed, with 91% of radicals scavenged at a concentration of 0.16% *w*/*v* and an estimated IC_50_ value of around 0.04% *w*/*v* as shown in Figure 4d. As in the case of the dry milled sample, most of the activity is associated with the particles themselves, as after centrifugation the supernatant scavenges only around 20% of radicals, whereas the resuspended pellet containing the particles neutralizes 89%, as illustrated in Figure 4e and f, respectively. This relation between supernatant and particle does not change much over a 24 h period, possibly due to the fact that liberation of soluble materials was largely accomplished already during the harsh conditions and elevated temperatures associated with wet milling, as shown in Appendix A.

The activity of an ethanolic extract was also assessed in this assay for comparison, and as expected, a considerable radical scavenging activity was also observed, with around 80% of DPPH radicals scavenged at a concentration of 1.25 mg·mL^−1^ and an estimated IC_50_ of 0.8 mg·mL^−1^, as shown in Appendix A. Notably, although 1 mg·mL^−1^ formally corresponds to 0.1% *w*/*v*, and a direct comparison of the suspensions with the extract may therefore suggest comparable activities, the nanosuspension is composed of essentially insoluble and complex particles rich in glucans, chitin, melatonin and fibers, whereas the extract is a concentrated mixture of compounds soluble in ethanol and therefore also mostly in water. It is therefore rather astonishing that the suspensions show a comparable activity to the extract in this assay.

#### 3.2.2. FRAP Capacity

In contrast to the DPPH radical scavenging assay, the FRAP assay measures the ability of compounds to reduce Fe^3+^ to Fe^2+^ by one electron transfer. The antioxidant power measured in these different assays may therefore occasionally also differ depending on the reactivity of the antioxidant substances in question. Interestingly, the suspensions of *Fomes fomentarius* were very active in the FRAP assay, reducing 93% and 50% of Fe^3+^ to Fe^2+^ at concentrations of 0.04% *w*/*v* in the dry milled and wet milled sample, respectively, as shown in Figure 5. Notably, the dry milled sample is more active in this assay and, as before, the main activity resides in the particles and not in the supernatant, with 58% reducing power in the case of the dry milled suspended and subsequently pelleted and resuspended particles versus 27% in the respective supernatant, and 50% reducing power in the particles versus just 10% in the supernatant for the wet milled sample, Figure 5b and c and e and f, respectively. Interestingly, stirring the suspension for 24 h did not result in any major increases in activity in the supernatants or indeed decreases in activity in the particles, counting against any further diffusion of FRAP active materials out of the particles into solution, as shown in Appendix A, an interesting issue which will be discussed later on. For comparison, the ethanolic extract at a formally similar concentration of 0.32 mg·mL^−1^ reduces around 40% of Fe^2+^ to Fe^3+^, as shown in Appendix A, highlighting again the considerable activity of the porous microparticles in these antioxidant assays.

#### 3.2.3. ABTS Capacity

In order to complement the antioxidant assays with another independent radical based assay, the ABTS assay is based on the reduction of the ABTS^●+^ radicals, itwas employed and the data is shown in Figure 6 [25]. As in the DPPH and FRAP assays, the suspensions of the dry milled and also wet milled samples of *Fomes fomenatrius* are both active, in a concentration dependent manner, and scavenging 81% and 90% of the ABTS radicals when employed at a concentration of 0.08% *w*/*v*, respectively. In this assay, the wet milled sample is more active compared to the dry milled one, with a very low IC_50_ value of around 0.005% *w*/*v*, compared to 0.04% *w*/*v* for the dry milled sample, a value comparable to the one also found in the other antioxidant assays. As in the DPPH and FRAP assays, the particles themselves are more active than the aqueous media they are suspended in, with 69% activity associated with the particles and 47% activity with the supernatant in the case of the dry milled sample and 64% activity associated with the pellet and 50% activity with the supernatant in the case of the wet milled sample. Intriguingly, in stark contrast to DPPH and in line with FRAP assay, stirring the suspension for 24 h did not result in any major increases in activity in the supernatants or indeed decreases in activity in the particles, counting against any further diffusion of active materials out of the particles into solution, as shown in Appendix A. The ethanolic extract at a comparable concentration of 0.32 mg·mL^−1^ neutralizes around 30% of the ABTS^●+^ radicals and at a concentration of 0.64 mg·mL^−1^ 50% of the radicals as shown in Appendix A.

### 3.3. Nematode Assay

The various antioxidant assays were complemented by a nematode assay based on the agricultural nematode *Steinernema feltiae.* Unlike the antioxidant tests *in vitro*, this assay considers the activity of the samples on a small multicellular organism and is therefore also indicative of (cyto-)toxicity. The data for a nominal concentration of 0.01% *w*/*v* are shown in Figure 7. Although the activities are more difficult to measure in such a complex assay, there is a general nematicidal activity associated with the dry and wet milled suspensions of *Fomes fomentarius* at 0.01% *w*/*v* after 24 h which is almost identical and again is primarily associated with the particles themselves and not so much with the aqueous medium, which is almost inactive in this assay. There may be a range of possible explanations for these findings, not necessarily related to redox activity, and these issues will be discussed later on. Notably the activity of the ethanolic extract at a concentration of 100 μg·mL^−1^ is also similar.

### 3.4. Qualitative Analysis of the Sample for Classes of Active Ingredients

As mentioned already, the fruiting body of the tinder conk, like most other sponges feeding from trees, is composed primarily of polymeric sugars, such as glucans and chitin, and also contains a number of additional, often redox active ingredients such as melatonin, uronic acid, glucuronic acid, 3,4-DHBA and purpurogallin. In order to perform a very preliminary assessment of the possible components responsible for the redox activities and (cyto-)toxicity observed in the various assays, a range of qualitative tests indicative of the relevant classes of compounds were performed, with the outcome summarized in Table 1. In short, these qualitative color tests confirm that the samples contain reactive and also reducing carbohydrates, alkaloids, glycosides and steroids. It was also possible to estimate the redox power of these particles based on gallic acid equivalents, and here the ethanolic extract at a concentration of 1 mg·mL^−1^, dry and wet suspensions at 0.8% *w*/*v* are equivalent to 30 µg·mL^−1^, 37.97 µg·mL^−1^ and 33.57 µg·mL^−1^ of gallic acid, respectively.

### 3.5. Cyclic Voltammetry

Apart from the various antioxidant assays, another way to determine possible antioxidant behavior of a given sample is based on an electrochemical investigation, such as Polarography or Cyclic Voltammetry (CV). Indeed, CV is often employed to explore complex mixtures of substances for redox activity and to obtain oxidation and reduction waves. As shown in Figure 8, the suspensions of dry and wet milled *Fomes fomentarius*, albeit clearly constituting a very complex system of numerous soluble and insoluble substances, show some redox activity. The values of anodic peak potentials (*E*_pa_) for 3,4-DHBA, purpurogallin, gallic acid, the suspension and the extract were recorded as 542.0 mV and 975.0 mV, 154.0 mV, 498.0 mV, 273.6 mV and 315.6 mV *vs* the Ag/AgCl reference electrode, respectively. Moreover, the values of the cathodic peak potentials (*E*_pc_) for 3,4-DHBA, purpurogallin, gallic acid, the suspension and the extract were recorded as −29.0, −732.0, −711.0, −728.0 and −657.6 mV *vs* the Ag/AgCl reference electrode, respectively, as shown, in Figure 8a and b.

Some of the more prominent redox active ingredients previously associated with *Fomes fomentarius* are based on *ortho*-quinones with their unique redox signals, as for instance 3,4-DHBA and purpurogallin, and their respective Cyclic Voltammogrammes are shown in Figure 8c,d. 3,4-DHBA, for instance, exhibits a quasi-reversible redox behavior with a reduction peak *E*_pc_ at −29.0 mV. This particular compound is oxidized to 4-formyl-1,2-benzoquinone in a two-electron step, with a first oxidation peak *E*_pa1_ at 542.0 mV and a second oxidation peak *E*_pa2_ at 975.0 mV. The values of charge of the transferred electrons (Q) were calculated based on the values of anodic peak potentials (*E*_pa_) for 3,4-DHBA, gallic acid, purpurogallin, ethanolic extract and wet suspension and values of 23.7 µC, 22.8 µC, 6.8 µC, 5.5 µC and 4.3 µC were obtained, respectively (Table 2).

## 4. Discussion

The studies conducted on the fruiting bodies of *Fomes fomentarius* have confirmed that it is possible to employ methods of size reduction, and here especially one step ball milling, to produce biologically active suspensions with particles with sizes in the higher nanometer and micrometer range. In this case, micro- or even nanosizing is possible as the dried tinder sponge is rather hard and brittle and hence amenable to milling, as are other plant derived materials, such as barks, shells, seeds, dried roots and pits [10,26,27]. The quality of particles produced by this milling method, of course, is not as good as the one of particles obtained by self-assembly, biology or precipitation [28,29,30]. Nonetheless, the preparation is straight forward, there is no need for complicated procedures or stabilizers, and the resulting suspensions, albeit prone to sedimentation in the absence of surfactants, are quite stable in the context of fouling, as ball milling is in itself an exothermic process and hence to some extent able to sterilize the samples and hence to prevent foul play [8,9]. Interestingly, ball milling is also possible in the absence of solvents, and the dry milled samples are especially durable and circumvent many of the issues associated previously with such nanosized suspensions, namely the persistent need for lyophilization and resuspension (NaLyRe) in the case of prolonged storage, transport and handling [29].

The antioxidant activity associated with these nanosuspensions is quite impressive and compares favorably to the ethanolic extract, which in any case is more tedious to produce and requires several steps involving large volumes of solvents, evaporation and solution in water. This antioxidant behavior is also noted by CV, on the one side with comparably low *E_pa_* values representative of an ease of electron donation and on the other side high Q values standing for a large amount of electrons donated.

The activity in the DPPH, FRAP and ABTS assays at first is primarily associated with the particles themselves and not with the surrounding medium, as after centrifugation, most of the activity stays with the particles. Despite prolonged periods of *de facto* maceration, there is no complete dissolution of antioxidant activity into the supernatant. In the case of the dry sample, exposure to an aqueous medium for 24 h obviously results in the liberation of some soluble antioxidants into the supernatant. This diffusion into the supernatant after 24 h is not as obvious in the wet milled samples, as these samples were milled in aqueous media and extraction by milling may have occurred already then.

In fact, the antioxidant activity associated with the particles themselves is rather interesting and likely to be due to the presence of insoluble, polymeric substances, such as the sugars caught by the Fehling’s and Benedict’s tests, which are obviously rather reducing [31,32]. These quite reducing sugars, which compose a large part of the sponge, cannot diffuse into the supernatant and rely on a specific surface reactivity. This surface reactivity is available in the dry and wet milled samples as size reduction in this already porous material results in vast surface areas. These sugary surfaces may react especially in the FRAP assay, explaining the high activity of the particles in this redox assay. In contrast, the DPPH assay is more focused on radical scavengers, and hence considerably higher IC_50_ values are found in the DPPH assay, and slow diffusion into the aqueous supernatant by such soluble radical scavenging agents is also more important, as witnessed in the dry sample stirred in an aqueous suspension for 24 h. Importantly, better diffusion of soluble materials out of the particles and into the supernatant also benefits from the milling process, as larger surface areas drive diffusion, as summarized in the Noyes-Whitney equation [33,34,35]. This may also explain that diffusion of reactive substances into the supernatant is more applicable in the DPPH assay, as this assay is based on more soluble radical scavenging compounds, such as quinones and gallates, for instance 3,4-DHBA and purpurogallin. Some studies have therefore assumed that nanosizing is just another way of fostering release of active ingredients, for instance from seeds and dried peels [8,36]. Although this may be correct in the case of materials rich in substances soluble in aqueous media, such as flavonoids, this is not necessarily the case in the case of the tinder sponge rich in reducing polymeric and insoluble sugars and similar insoluble substances weathering the harsh and often wet conditions in the forest [37,38]. Here, milling results in a significantly increased surface area of an already highly porous material, and therefore enables a pronounced reactivity at and activity associated with the surface of these particles which is, once again, especially large in the case of this porous material.

One should also note that such small, porous, solid and insoluble particles may also possess a specific physical activity. In the case of the nematodes, the (cyto-)toxicity recorded in this assay may therefore be due to a combination of actions. As the supernatant is as good as inactive against *Steinernema feltiae*, it is unlikely that any toxic substances are liberated. The activity of the pelleted and resuspended material is probably due to physical actions of these small insoluble chunks which may impact on the nematode in concert with their surface reactivity discussed before.

It is also noteworthy that dry milling on occasion results in comparable or even higher activity compared to wet milling. These difference are possibly due to a more rapid oxidation in aerated aqueous suspensions and at the elevated temperatures associated with the method of wet milling, once more underlining the elegance of the ball milling process as a versatile method to unleash the antioxidant activity of such materials also under dry and milder circumstances [8,9]. In any case, milling increases the surface area considerably, and therefore also increases the reactivity on or at surfaces. It also gives the chance for the diffusion of reactive substances from the sponge into the supernatant and for the diffusion of reactive substances from the supernatant into the sponge. Compounds attracted into the sponge may arrive there in high concentrations in the pores as these *de facto* are filled rapidly with aqueous media adsorbed into the sponge. It is therefore likely that DPPH, FRAP and ABTS are actually adsorbed into the particles and may actually enrich there, dependent on interactions with the surfaces inside the pores.

*Fomes fomentarius* has diverse medicinal applications including anti-bacterial, anti-diabetes, anti-inflammatory and antioxidant activities [39,40]. As dry milling produces a fine powder which can be stored for months and also easily transported, practical applications of these materials are not farfetched, for instance as wound dressings, antioxidant food supplements, agricultural fertilizers and plant protectants, cosmetics, sunscreens, and possibly even as materials, for instance as redox filters in the treatment of oxidizing waste (Figure 9). In some of these applications, the ability of the fruiting body of the dried tinder conk to adsorb and also absorb liquids may be beneficial, similar to uses of dried corncobs as effective eco-friendly absorbants derived from agricultural waste [41,42,43]. These aspects of size reduction, increased surface areas and reactivities and also the biological activities and possible applications associated with the suspensions of *Fomes fomentarius* clearly require further and more in-depth investigations.

## 5. Conclusions

In the case of *Fomes fomentarius*, size reduction and suspension, either directly by wet milling or via dry milling and subsequent suspending in aqueous media, grants access to a considerable antioxidant and biological activity associated with this natural redox active material, and without the need of solvent extraction and preparation. As the tinder sponge itself is readily available in many woodlands and harvesting its fruiting body is straightforward and does not destroy the host or eco-system, this fungus could be of considerable interest for many practical and often eco-friendly applications. Further studies should therefore address the optimization of the milling procedure, possibly reducing the size further into the nanorange and stabilizing the suspension, for instance by adding a surfactant, by converting it into crèmes for topical applications, mixtures for food, gels for agricultural applications, or fixed supports for applications in the treatment or filtering of wastewater. The mode of action, especially the slow liberation of active substances by diffusion and also during biodegradation of the material should be studied, as should the stability and biodegradation itself. Interestingly, the sponge itself possesses a large surface area already, and milling such spongy material potentiates this specific reactivity at or near such surfaces in the pores. These issues are unique to such spongy materials and worth further investigation in the context of surface reactivity, ad- and absorption and also in the context of attaching reactive substances, such as flavonoids to the surface of these materials and hence changing this already active material into a carrier for more or different activity and action.

## Figures and Tables

**Figure 1 antioxidants-10-00303-f001:**
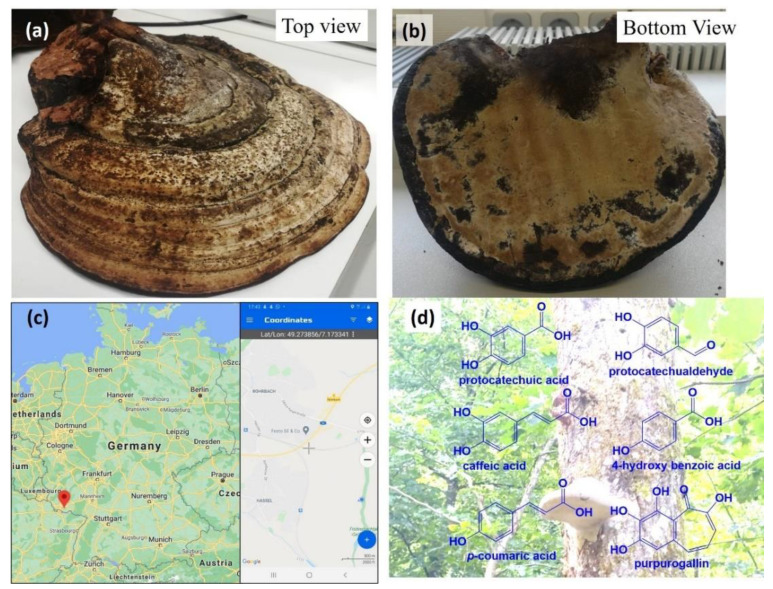
A fine specimen of the fruiting body of tinder conk *Fomes Fomentarius* weighing 2122.17 g before and 826.68 g after drying as shown in (**a**) and (**b**) was collected in the woodland surrounding the town of Hassel/Saar in Saarland, Germany, as highlighted on the map of Germany in (**c**) (credit Google Maps). Its botanical identity was determined by Dr Martin C. H. Gruhlke at the Department of Plant Physiology at RWTH Aachen, Aachen, North Rhine-Westphalia and the samples of sponge were processed further as part of this study. A section of active ingredients commonly found in such sponges as described in the literature is shown in (**d**) [4,14].

**Figure 2 antioxidants-10-00303-f002:**
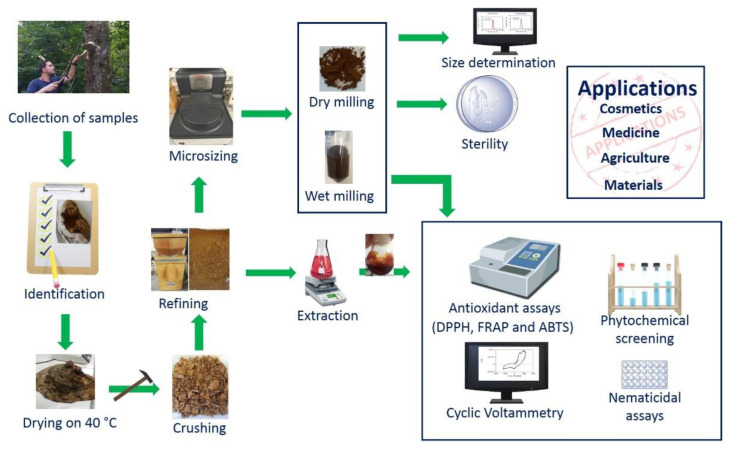
Illustration of the procedures and materials involved in this study. It should be emphasized that no trees were harmed during the collection of the samples. The photos of the harvest and materials are kindly provided by Muhammad Jawad Nasim and Abdul Karim Darkal who is also depicted harvesting a fine specimen of *Fomes fomentarius*.

**Figure 3 antioxidants-10-00303-f003:**
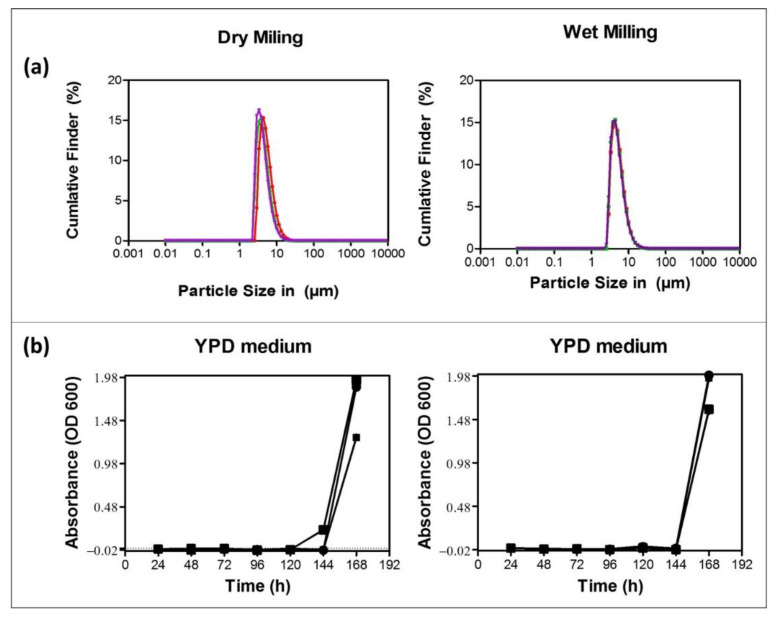
Physical and microbial analysis of the dry milled powders and wet milled suspensions confirm small particles with diameters in the range of 1 to 10 µm, as shown in (**a**), and a notable sterility of these samples in yeast peptone dextrose (YPD) media for up to six days as indicated in (**b**). Similar results were obtained once lysogeny broth (LB) medium was used rather than YPD medium.

**Figure 4 antioxidants-10-00303-f004:**
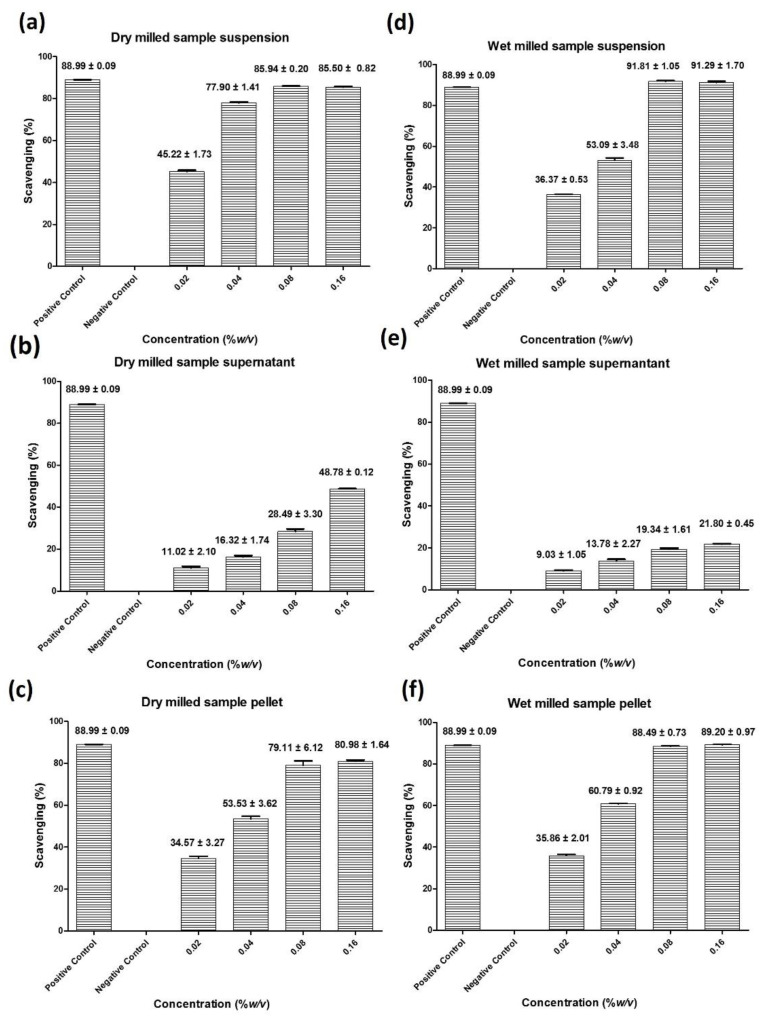
The dry and wet milled samples in suspension scavenge the DPPH radical efficiently. This radical scavenging ability at first is associated primarily albeit not exclusively with the particles themselves, counting towards a specific surface reactivity of the highly porous microparticles and in general against a quick diffusion of active ingredients into the surrounding aqueous medium, especially in the case of the wet milled suspension. Dry milled and suspended sample (**a**), supernatant after centrifugation at 19,000× *g* (**b**), resuspended pellet of particles (**c**), wet milled suspension (**d**), supernatant after centrifugation (**e**), resuspended pellet of particles (**f**). Values represent the mean ± SD (*n* = 3).

**Figure 5 antioxidants-10-00303-f005:**
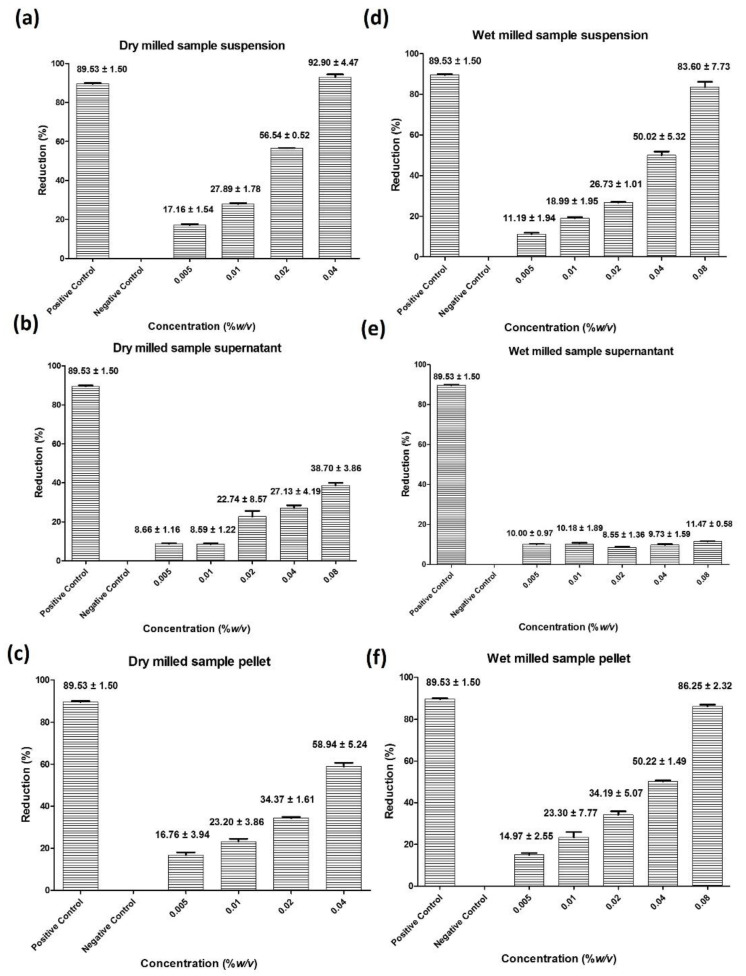
Suspensions of dry and wet milled *Fomes fomentarius* reduce Fe^3+^ ions to Fe^2+^ ions in the ferric reducing antioxidant potential (FRAP) assay. This reducing ability at first is associated primarily albeit not exclusively with the particles themselves, counting towards a specific surface reactivity of the highly porous microparticles and in general against a quick diffusion of active ingredients into the surrounding aqueous medium, especially in the case of the wet milled suspension. Dry milled and suspended sample (**a**), supernatant after centrifugation at 19,000× *g* (**b**), resuspended pellet of particles (**c**), wet milled suspension (**d**), supernatant after centrifugation (**e**), resuspended pellet of particles (**f**). Values represent the mean ± SD (*n* = 3).

**Figure 6 antioxidants-10-00303-f006:**
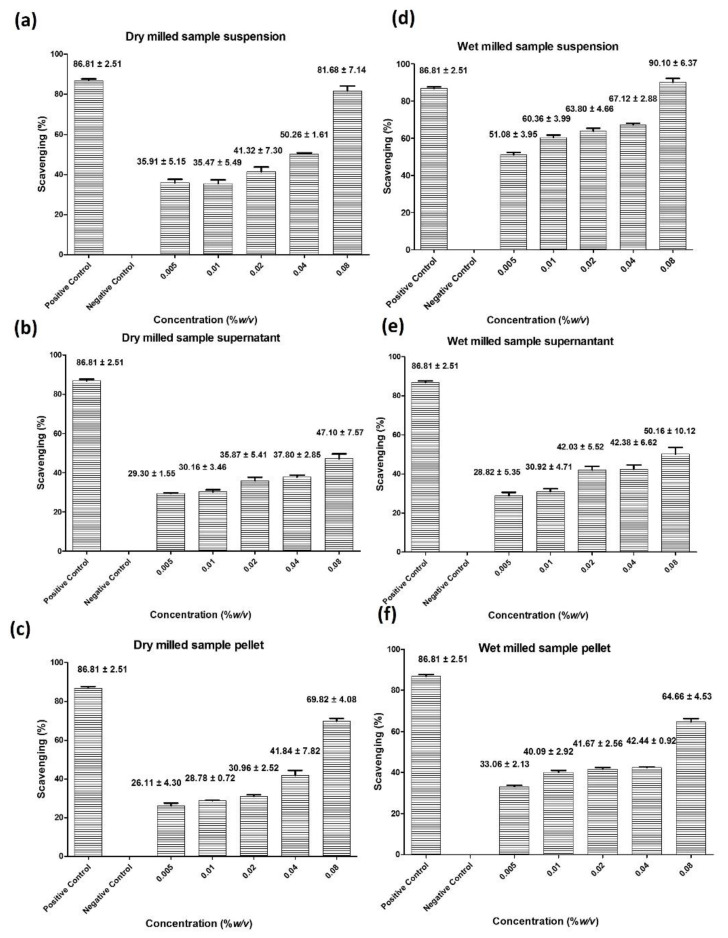
Suspensions of dry and wet milled *Fomes fomentarius* reduce 2,2′-azino-*bis*(3-ethylbenzothiazoline-6-sulfonic acid (ABTS^●+^) radicals. This reducing ability at first is associated primarily albeit not exclusively with the particles themselves, counting towards a specific surface reactivity of the highly porous microparticles and in general against a quick diffusion of active ingredients into the surrounding aqueous medium, especially in the case of the wet milled suspension. Dry milled and suspended sample (**a**), supernatant after centrifugation at 19,000× *g* (**b**), resuspended pellet of particles (**c**), wet milled suspension (**d**), supernatant after centrifugation (**e**), resuspended pellet of particles (**f**). Values represent the mean ± SD (*n* = 3).

**Figure 7 antioxidants-10-00303-f007:**
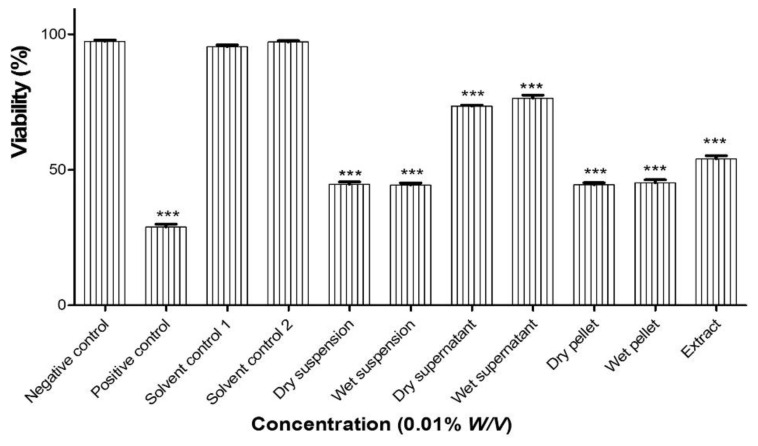
Activity of the dry and wet milled suspensions against the agricultural nematode *Steinernema feltiae*. Although these data are more difficult to assess due to the complexity of this in vivo assay, the dry and the wet milled samples are rather active, with almost identical activities which are clearly associated especially with the particles themselves and also comparable to the one of the ethanolic extract at a concentration of 100 μg·mL^−1^. Values represent the mean ± SD *** *p* < 0.05 (*n* = 9).

**Figure 8 antioxidants-10-00303-f008:**
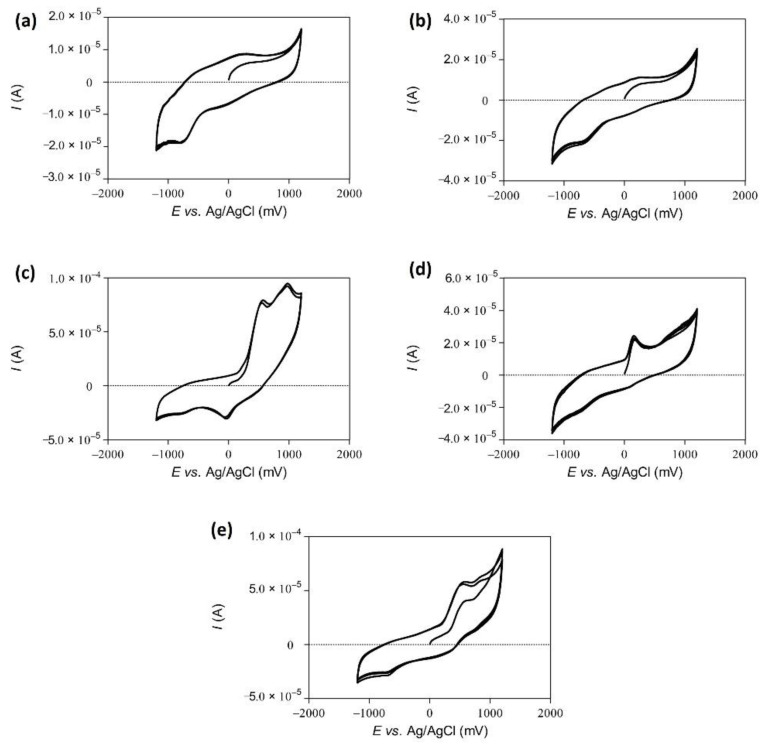
Cyclic Voltammograms of the wet milled suspension, (**a**), ethanolic extract (**b**), and reference compounds present in such sponges, purpurogallin (**c**) and 3,4-DHBA (**d**). Gallic acid under the same experimental conditions is shown in (**e**). Experimental details are provided in the text.

**Figure 9 antioxidants-10-00303-f009:**
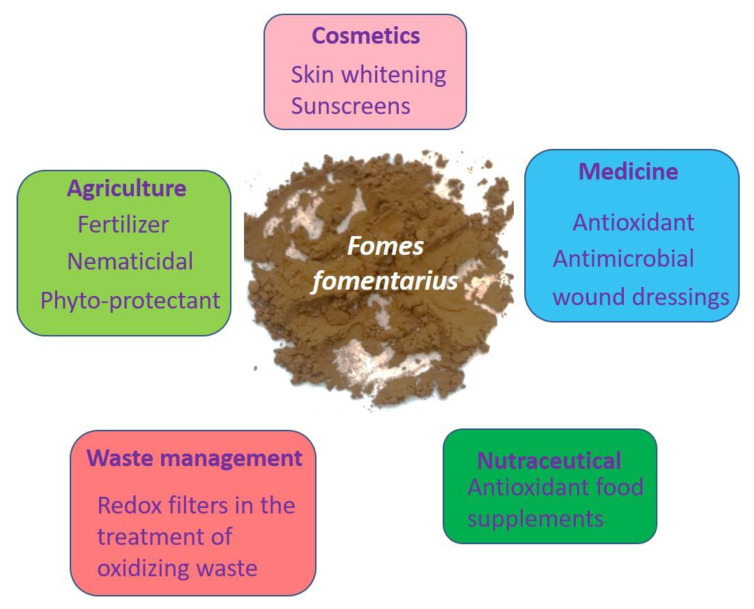
A few selected possible applications associated with *Fomes fomentarius* from medicine and nutrition to agriculture, materials and cosmetics.

**Table 1 antioxidants-10-00303-t001:** Qualitative assessment of the composition and reactive ingredients which may in part be responsible for the antioxidant and (cyto-)toxic activities observed. A + sign indicates the presence and the − sign indicates the absence of phytochemicals.

Class of Compounds	Assays	Dry Suspension	Wet Suspension	Extract
Carbohydrates	Molisch’s Test	+	+	+
	Fehling’s Test	+	+	+
	Benedict’s Test	+	+	+
Alkaloids	Wagner’s Test	+	+	+
	Mayer’s Test	−	−	−
	Dragendorff’s Test	+	+	+
Glycosides	Foam Test	+	+	+
	Legal’s Test	+	+	+
	Keller-Killiani Test	−	−	+
Steroids	Salkowski’s Test	+	+	+

**Table 2 antioxidants-10-00303-t002:** Data obtained from Cyclic Voltammetry confirms a reducing wave in the suspension and extracts with *E*_pa_ values of around 274 mV and 316 mV versus Ag/AgCl, respectively. The respective values for 3,4-DHBA and purpurogallin, compounds present in such sponges, are provided for comparison.

Samples	*E*_pa1_ (mV)	*I*_pa1_ (µA)	Q(µC)	*E*_pa2_ (mV)	*I*_pa2_ (µA)	*E*_pc_ (mV)	*I*_pc_ (µA)
Wet suspension	273.6	9.6	4.3	-	-	−728.0	−19.4
Extract	315.6	17.3	5.5	-	-	−657.6	−26.4
3,4-DHBA	542.0	63.3	23.7	975.0	61.6	−29.0	−36.2
Purpurogallin	154.0	22.6	6.8	-	-	−732.0	−37.1
Gallic acid	498.0	57.1	22.8	-	-	−711.0	−26.8

## Data Availability

The data presented in this study are available in the article and Appendix A.

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
