# Peer review of "Unleashing the Biological Potential of Fomes fomentarius via Dry and Wet Milling"

_antioxidants, 2021, doi:10.3390/antiox10020303_

Round 1

Reviewer 1 Report

The manuscript is not suitable for publication in Antioxidants because of the following reasons:

  1. The title of the manuscript does not represent its content and aim.
  2. It is not necessary to mention the brand of the mil in the abstract “Dry or wet ball milling of this sponge 10 on a “Pulversisette 7 premium line” planetary mill results in small particles with sizes in the range 11 of 10 μm or below”, because similar particle size could be probably achieved with other mills, as well.
  3. Abstract: This is speculation “probably due to the increased temperatures of around 80°C, especially during the wet”, which was not proven by the authors and should be removed from the abstract.
  4. What is the value of figure 1? The information bellow the figure could be presented in materials and methods section and panel D in the introduction part. Moreover, in its current form, the same information for the sourcing and identification is repeating.
  5. Authors should justify the choice of 98% ethanol as extracting agent. What compounds do they want to extract with this extragent? It is know that water-ethanol mixtures (60-70%) are far better extragents of the majority of polyphenols in comparison to 98% ethanol.
  6. This text is speculation “….. and temperatures in the chamber can reach 80 oC. The resulting powders and suspensions, respectively, thus have been automoatically sterilized by the milling process and have turned out as rather resistant to microbial contamination, with no significant fouling for up to six days, as illustrated in Figure 3.”, which is not proved by the authors. Moreover, the temperature of the dry milling reached only 41.6°C. Five minutes treatment at such low temperature are not enough for suitable sterilization.
  7. Used tests for antioxidant activity are not physiologically relevant. Moreover, the expression of the results as % inhibition is not informative as their expression in equivalents of a known antioxidant (i.e. Trolox, gallic acid, ascorbic acid, etc). In addition, expressions such as “The antioxidant activity associated with these nanosuspensions is quite impressive” are inconsistent without being compared to compounds with known antioxidant activity or results from other studies.
  8. Figure 9 does not provide analytical results and should be presented as supplementary material.
  9. Probably the biggest weakness of the study is the use of spectrophotometric methods to measure absorbance of suspensions, which moreover are consisted of particles of different size, are not stabilized and prone to sedimentation. This could interfere with the recorded absorbance values and lead to distorted results.
  10. It is stated “The content of phenolic acids was quantified employing the protocol described in the literature”, however no data for the presence of phenolic acids in the suspensions is presented.
  11. The used Qualitative screening for different classes of substances is not informative enough. The analyses of total polyphenols and total flavonoids would be more useful in assessment the content of antioxidant in the samples.
  12. The study lacks statistical analysis.
  13. There are some technical and grammatical errors that should be corrected.

Reviewer 2 Report

Unleashing the tinder from Fomes fomentarius

I write you regarding manuscript # antioxidants-1080649 entitled "Unleashing the tinder from Fomes fomentarius," which you submitted to the antioxidants-1080649.

The authors need to follow the following instructions to improve this manuscript.

  • The author should justify the title, objectives and findings
  • Page 1, Line 9-21 (Abstract section): Please, insert some relevant numeric result more; and write the best findings
  • Page 2, Line 67-69: exactly mention the room temperature.
  • Page 2, Figure 1(d): Is this identified by the author? Otherwise, add the reference.
  • Page 3, Figure 2: (Application) Is it agriculture or Food?
  • In this manuscript: Please change rpm to g value if possible
  • Page 3, Line 90: (25 %w/v) Check the space
  • Page 7, Line 231: Check the full stop
  • Page 7, Line 235: DPPH assay change to DPPH capacity
  • Page 8, Line 281: FRAP assay change to FRAP capacity
  • Page 10, Line 309: ABTS assay change to ABTS capacity
  • In the manuscript, numerical value and % write without space
  • In the figure, for example, 6, showed numerical value and %. I think the author mentioned in left alignment (Scavenging %). So, there is no need to write double %. Just show 86.81, 35.91, ----. Follow all figures.
  • In figure 7, show the numerical value.
  • In the entire manuscript, the author should mention the replication number.
  • In the figure, they mentioned the bar but did not mention deviation/error.
  • Page 18, Line 486-505: Conclusions part should improve with the best findings.
  • English Grammar should check by a Native English Speaker or a commercial proofreading company.
  • References carefully check before resubmission.

I recommend to improve the manuscript and resubmit.

Author Response

Reviewer 2

  • The author should justify the title, objectives and findings

Response: The title, objectives and findings have been edited.

  • Page 1, Line 9-21 (Abstract section): Please, insert some relevant numeric result more; and write the best findings

Response: We thank the reviewer for this suggestion. The numeric values have been added to the abstract.

  • Page 2, Line 67-69: exactly mention the room temperature.

Response: We thank the reviewer for the critical reading. The samples were not dried at room temperature but in oven at 40 °C. This information has been corrected in the manuscript.

  • Page 2, Figure 1(d): Is this identified by the author? Otherwise, add the reference.

Response: we thank the reviewer for this suggestion. The references have been added to the figure legend.

  • Page 3, Figure 2: (Application) Is it agriculture or Food?

Response: Since this mushroom is inedible, we considered the potential application of this mushroom in the field of agriculture as phyto-protectant.

  • In this manuscript: Please change rpm to g value if possible

Response: RPM values have been converted to g values throughout the manuscript.

  • Page 3, Line 90: (25 %w/v) Check the space

Response: It has been edited.

  • Page 7, Line 231: Check the full stop

Response: It has been edited.

  • Page 7, Line 235: DPPH assay change to DPPH capacity

Response: It has been edited.

  • Page 8, Line 281: FRAP assay change to FRAP capacity

Response: It has been edited.

  • Page 10, Line 309: ABTS assay change to ABTS capacity

Response: It has been edited.

  • In the manuscript, numerical value and % write without space

Response: We thank the reviewer for this valuable comment. We have edited the text accordingly.

  • In the figure, for example, 6, showed numerical value and %. I think the author mentioned in left alignment (Scavenging %). So, there is no need to write double %. Just show 86.81, 35.91, ----. Follow all figures.

Response: We thank the reviewer for this valuable comment. We have edited the figures accordingly.

  • In figure 7, show the numerical value.

Response: We thank the reviewer for this valuable comment. We have edited the figures accordingly.

  • In the entire manuscript, the author should mention the replication number.

Response: The manuscript has been edited accordingly.

In vitro assays were performed as three different repetition n = 3 , while in vivo assays were performed as three repetition in three different occasion n = 9.

  • In the figure, they mentioned the bar but did not mention deviation/error.

Response: The figures have been edited accordingly.

  • Page 18, Line 486-505: Conclusions part should improve with the best findings.

Response: The best findings are provided in the Results section and discussed in the Discussion in a very short and direct manner, hence it would be repetitive to repeat them again and inflate the Conclusions which are also kept as short and informative as possible

  • English Grammar should check by a Native English Speaker or a commercial proofreading company.

Response: The manuscript has been written by a native speaker holding a second degree in Philosophy and former senior lecturer at the University of Exeter in the UK.

  • References carefully check before resubmission.

Response: We have checked the refences carefully.

I recommend to improve the manuscript and resubmit.

Reviewer 3 Report

The work is of general interest, is well planned and described. In my opinion the paper is worth studying and the manuscript contains enough original material. The experimental tests are carried out correctly using appropriate methods. The results are quite interesting and well statistically analyzed.

Minor corrections:

Text formatting should be carefully checked.

The language should be modified carefully.

Author Response

Reviewer 3

The work is of general interest, is well planned and described. In my opinion the paper is worth studying and the manuscript contains enough original material. The experimental tests are carried out correctly using appropriate methods. The results are quite interesting and well statistically analyzed.

Minor corrections:

Text formatting should be carefully checked.

The language should be modified carefully.

Response: We thank the reviewer for the encouraging remarks. The manuscript is written by a native speaker and he has checked it again to polish the language and to carry out grammatical and formatting changes.

Reviewer 4 Report

  1. Regarding the name of the investigated biological material, the authors used both the scientific (Fomes fomentarius) and the common (tinder conk) names one after another (line 9, 34). A more appropriate wording would be "Fomes fomentarius (commonly known as the tinder conk)".
  2. The way the sentences at lines 34 and 35 are formulated makes the meaning unclear.
  3. The paragraph between lines 40-44 is too long and difficult to be understood in its present form.
  4. Line 58: the form of the verb ("haves") neither exists nor is it appropriate to the subject of the sentence.
  5. For Chapter 2. Materials and Methods, I would recommend the authors include a separate subchapter describing all the reagents (including culture media) used for their experiments.
  6. Regarding the evaluation of possible microbial contamination, the authors used two culture media appropriate to growth one Gram negative bacteria ( E.coli) – LB (which type of LB medium was used?) and a yeast (S. cerevisiae) - YPD.  Why did the authors focus on testing for possible contamination with only these 2 microbial species? Literature data show that fruiting bodies of Fomes fomentarius could be in contact with the black tinder fungus beetle Bolitophagus reticulatus, a fungivorous species occurring widely throughout European forests - and various bacterial communities associated with larvae and adults of this species can be present. Authors should take this aspect into consideration in order to correctly assess the microbial contamination of the investigated samples.
  7. The authors state that sterile distilled water was used to prepare the wet-milled sample. Was the sterile distilled water used to prepare the suspension of the dry milled sample, and also to re-suspend the pellet?
  8. Regarding the sterilization process associated with the milling procedure by reaching 800C (only for the wet-milled sample as described in subchapter 2.2 because the dry-milled procedure resulted in a temperature of 41.6 0C) this statement is true for most bacteria and fungi, however the sporulated forms are destroyed at higher temperatures (e.g. by dry heat for 60 minutes at 120-160 degrees Celsius for bacteria, and over 115 degrees Celsius for fungi, and in humid heat those several spores are destroyed in 30 minutes at over 120 degrees Celsius).
  9. Figures 1-6 presented in the Supplementary Material should be included as Figure 1, Figure 2, etc; in the text of the manuscript at "see Supplementary Material".
  10. For DPPH and ABTS assays, the authors presented in figures 4 and 6 the results comparatively between dry and wet milled samples of the same concentrations. This is not valid for FRAP assay as in figure 5 the effect produced by the dry milled sample with the concentration of 0.08% is not shown. The authors should explain this issue.
  11. At 3. Nematode assay, which is the correct concentration of the Fomes fomentarius sample which was tested, 0.1% w/v (line 343) or 0.01% w/v as was indicated in Figure 7?
  12. Regarding the active ingredients found in the tinder conk's fruiting body, the authors state that "uronic acid, glucuronic acid" are present. The authors should specify which uronic acid is present because uronic acids are a class of compounds comprising different representatives (e.g. glucuronic acid, galacturonic acid, etc).
  13. An interesting application is the possibility to store the fine powder resulted from the milling procedures for long periods ("for months" – line 472) to be used for different practical applications. Did the authors test the possibility of storing these samples from Fomes fomentarius for measured extended periods of time, while observing that the samples maintain all the properties discussed in such a way that they can be used in optimal conditions in practice? Other important applications that should be mention are in various other fields of medicine because of numerous additional important bioactivities (e.g. antidiabetic, antitumor, anti-inflammatory, antiviral, antiatherogenic, etc.) which are described in literature.
  14. Some minor language and typing errors: line 196: I suggest the use of "further to be presented" instead of "now be presented; line 219: "automoatically" to be corrected.

Author Response

Reviewer 4

  1. Regarding the name of the investigated biological material, the authors used both the scientific (Fomes fomentarius) and the common (tinder conk) names one after another (line 9, 34). A more appropriate wording would be "Fomes fomentarius (commonly known as the tinder conk)".

Response: We thanks the reviewer for this suggestion and have edited the text accordingly.

  1. The way the sentences at lines 34 and 35 are formulated makes the meaning unclear.

Response: The sentence has been edited.

  1. The paragraph between lines 40-44 is too long and difficult to be understood in its present form.

Response: We have divided the paragraph into several sentences and edited the paragraph to make it clearer.

  1. Line 58: the form of the verb ("haves") neither exists nor is it appropriate to the subject of the sentence.

Response: We thank the reviewer for reading of the manuscript so nothing has escaped her or his inspection. The text has been edited.

  1. For Chapter 2. Materials and Methods, I would recommend the authors include a separate subchapter describing all the reagents (including culture media) used for their experiments.

Response: We thank the reviewer for the valuable suggestion. We have added a subsection about the reagents.

  1. Regarding the evaluation of possible microbial contamination, the authors used two culture media appropriate to growth one Gram negative bacteria (coli) – LB (which type of LB medium was used?) and a yeast (S. cerevisiae) - YPD. Why did the authors focus on testing for possible contamination with only these 2 microbial species? Literature data show that fruiting bodies of Fomes fomentarius could be in contact with the black tinder fungus beetle Bolitophagus reticulatus, a fungivorous species occurring widely throughout European forests - and various bacterial communities associated with larvae and adults of this species can be present. Authors should take this aspect into consideration in order to correctly assess the microbial contamination of the investigated samples.

Response:

Response: We thank the reviewer for the valuable comment. We agree with the reviewer that the sterility should been considered for the range of other possible contaminations. The aim of our study was to confirm the sterility of our sample for a few days in order to enable us to perform our tests, and we have done this by showing the absence of the most basic microorganisms via a very basic pre-screening. Indeed, contamination with microorganisms has been a very tricky matter with milling in our previous studies and we have been quite satisfied that the material did not foul within a few days.

  1. The authors state that sterile distilled water was used to prepare the wet-milled sample. Was the sterile distilled water used to prepare the suspension of the dry milled sample, and also to re-suspend the pellet?

Response: Yes the sterilized distilled water was used for preparing the suspension of dry milled sample and resuspending the pellet and this is now mentioned in the text.

  1. Regarding the sterilization process associated with the milling procedure by reaching 800C (only for the wet-milled sample as described in subchapter 2.2 because the dry-milled procedure resulted in a temperature of 41.6 0C) this statement is true for most bacteria and fungi, however the sporulated forms are destroyed at higher temperatures (e.g. by dry heat for 60 minutes at 120-160 degrees Celsius for bacteria, and over 115 degrees Celsius for fungi, and in humid heat those several spores are destroyed in 30 minutes at over 120 degrees Celsius).

Response: We completely agree with the reviewer that exposure to a high temperature for a brief period can not be considered enough for sterilization. The inherent antimicrobial potential of Fomes fomentarius should not be underestimated as described in literature:

https://www.sciencedirect.com/science/article/pii/S0926669015304702

  1. Figures 1-6 presented in the Supplementary Material should be included as Figure 1, Figure 2, etc; in the text of the manuscript at see Supplementary Material.

Response: We thank the reviewer for this valuable suggestion. We have edited the text accordingly.

  1. For DPPH and ABTS assays, the authors presented in figures 4 and 6 the results comparatively between dry and wet milled samples of the same concentrations. This is not valid for FRAP assay as in figure 5 the effect produced by the dry milled sample with the concentration of 0.08% is not shown. The authors should explain this issue.

Response: We thank the reviewer for raising this point. The FRAP assay was performed at 0.08% w/v concentration for both dry suspension and wet suspensions and provided the reduction of more than 100%. The sample concentration of 0.08% w/v was therefore excluded from the assay.

  1. At Nematode assay, which is the correct concentration of the Fomes fomentarius sample which was tested, 0.1% w/v (line 343) or 0.01% w/v as was indicated in Figure 7?

Response: The correct concentration is 0.01% w/v. It has been corrected in the manuscript.

  1. Regarding the active ingredients found in the tinder conk's fruiting body, the authors state that "uronic acid, glucuronic acid" are present. The authors should specify which uronic acid is present because uronic acids are a class of compounds comprising different representatives (g. glucuronic acid, galacturonic acid, etc).

Response:  We thank the reviewer for the critical reading. It has been corrected in the manuscript. 

  1. An interesting application is the possibility to store the fine powder resulted from the milling procedures for long periods ("for months" – line 472) to be used for different practical applications. Did the authors test the possibility of storing these samples from Fomes fomentarius for measured extended periods of time, while observing that the samples maintain all the properties discussed in such a way that they can be used in optimal conditions in practice?

Response: We thank the reviewer for this very important question. We have performed such a study on another sample (comprising of chalcogens) in the past using another technology. The details are described in the following manuscript:

Griffin, S.; Sarfraz, M.; Hartmann, S.F.; Pinnapireddy, S.R.; Nasim, M.J.; Bakowsky, U.; Keck, C.M.; Jacob, C. Resuspendable Powders of Lyophilized Chalcogen Particles with Activity against Microorganisms. Antioxidants 2018, 7, 23. https://doi.org/10.3390/antiox7020023

We are planning to look at this aspect as part of a long term stability stud and are likely to describe this as part of another manuscript.

  1. Other important applications that should be mention are in various other fields of medicine because of numerous additional important bioactivities (g. antidiabetic, antitumor, anti-inflammatory, antiviral, antiatherogenic, etc.) which are described in literature.

Response: We thank the reviewer for this very important comment. We have added a few possible medicinal applications in the text, although we always hesitate to provide such claims before we have conducted the studies necessary.

  1. Some minor language and typing errors: line 196: I suggest the use of "further to be presented" instead of "now be presented; line 219: automoatically to be corrected.

Response:

Response: These mistakes have been corrected as instructed.

Round 2

Reviewer 1 Report

Authors have addressed satisfactory most of reviewers' comments and remarks and now manuscript is suitable for publication.

Reviewer 2 Report

Accept the cleaned file